# Transcriptomic Analysis of Air–Liquid Interface Culture in Human Lung Organoids Reveals Regulators of Epithelial Differentiation

**DOI:** 10.3390/cells13231991

**Published:** 2024-12-02

**Authors:** Jieun Kim, Eun-Young Eo, Bokyong Kim, Heetak Lee, Jihoon Kim, Bon-Kyoung Koo, Hyung-Jun Kim, Sukki Cho, Jinho Kim, Young-Jae Cho

**Affiliations:** 1Division of Pulmonary and Critical Care Medicine, Department of Internal Medicine, Seoul National University College of Medicine, Seoul National University Bundang Hospital, Seongnam 13620, Republic of Korea; jekim2022@chauniv.ac.kr (J.K.); r0713@snubh.org (E.-Y.E.); r2980@snubh.org (B.K.); dr.hjkim@snubh.org (H.-J.K.); 2Department of Biomedical Science, CHA University, Seongnam 13488, Republic of Korea; 3Center for Genome Engineering, Institute for Basic Science, Daejeon 34126, Republic of Korea; leeheetak@ibs.re.kr (H.L.); jhkim@catholic.ac.kr (J.K.); koobk@ibs.re.kr (B.-K.K.); 4Department of Medical and Biological Sciences, The Catholic University of Korea, Bucheon 14662, Republic of Korea; 5Department of Thoracic and Cardiovascular Surgery, Seoul National University Bundang Hospital, Seongnam 13620, Republic of Korea; skcho@snubh.org; 6Department of Genomic Medicine, Seoul National University Bundang Hospital, Seongnam 13620, Republic of Korea; 7Precision Medicine Center, Future Innovation Research Division, Seoul National University Bundang Hospital, Seongnam 13620, Republic of Korea; 8Department of Laboratory Medicine, Seoul National University College of Medicine, Seoul National University Bundang Hospital, Seongnam 13620, Republic of Korea

**Keywords:** airway epithelium, air–liquid interface, ciliated cells, differentiation, single-cell RNA-sequencing

## Abstract

To develop in vitro respiratory models, it is crucial to identify the factors involved in epithelial cell differentiation. In this study, we comprehensively analyzed the effects of air–liquid interface (ALI) culture on epithelial cell differentiation using single-cell RNA sequencing (scRNA-seq). ALI culture induced a pronounced shift in cell composition, marked by a fivefold increase in ciliated cells and a reduction of more than half in basal cells. Transcriptional signatures associated with epithelial cell differentiation, analyzed using iPathwayGuide software, revealed the downregulation of *VEGFA* and upregulation of *CDKN1A* as key signals for epithelial differentiation. Our findings highlight the efficacy of the ALI culture for replicating the human lung airway epithelium and provide valuable insights into the crucial factors that influence human ciliated cell differentiation.

## 1. Introduction

The airway epithelium acts as a physical barrier that prevents pathogens, including inhaled particles and microbes, from entering the respiratory system [1]. It is primarily composed of ciliated, secretory, and basal cells. Ciliate cells protect respiratory health by expelling pathogens, which are surrounded by mucus produced by secretory cells from the respiratory tract [2,3]. Basal cells are primary airway stem cells that differentiate into secretory and ciliated cells to maintain homeostasis following injury [4]. Ciliated-cell differentiation for epithelial regeneration after injury can occur through the direct differentiation of basal cells or the transdifferentiation of secretory cells [1,2,4].

Human airway epithelial cells in culture are commonly used to study human lung development and to model respiratory diseases [5,6,7]. These cells can be obtained from lung tissue biopsies and are cultured as primary airway epithelial cells in airway 3D organoids or air–liquid interface (ALI) systems [6,7]. Airway 3D organoids derived from basal or club cells can be maintained in the long term and generate differentiated cells, allowing the study of cellular lineage hierarchy in the human airway epithelium, although they fail to contain ciliated cells [7,8]. In the ALI system, basal cells are in contact with the air and can undergo proper differentiation into a pseudostratified epithelium consisting of ciliated and secretory cells, providing a robust in vitro airway model. In this system, epithelial cells are cultured on a permeable membrane with a medium in the basal chamber, and the epithelium is exposed to air on the apical side of the membrane [9,10]. This configuration mimics the conditions found in the human airway and drives differentiation towards a mucociliary phenotype [2,9]. However, with only exploratory results, no consensus has been reached regarding the standardization of the ALI method, limiting its reliability and reproducibility [11]. Although it is essential for the in vitro modeling of respiratory diseases to include ciliated cells, which exist in high proportions in the human respiratory epithelium and are important as the major target of SARS-CoV-2, the ALI method does not produce enough ciliated cells, as observed in the human airway epithelium [9]. Because the generation of in vitro models containing a sufficient number of ciliated cells remains a challenge, even with the ALI system, identifying the factors involved in ciliated cell differentiation is important for developing a suitable in vitro airway epithelial model.

The purpose of this study was to compare the transcriptomic gene signatures of adult tissue-derived human lung organoids cultured using the ALI method to those cultured using the submerged method, using single-cell RNA sequencing (scRNA-seq), to identify the key regulatory signaling pathways involved in the ALI method that contribute to ciliated cell differentiation. We also discovered chemicals that modulate the activity of key regulators, which were found to replace the differentiation effect of the ALI method.

## 2. Materials and Methods

### 2.1. Organoid Culture

Primary lung organoids were generated from the fresh normal lung tissues of two de-identified patients undergoing tumor resection surgery at Seoul National University Bundang Hospital (SNUBH), with written informed consent and ethical approval provided (IRB No. B-1409-265-001). Neither donor (aged 55–60, smoker; aged 70–75, non-smoker) had a history of respiratory disease. Tissue samples were selected from the areas farthest from the tumor lesions, dissociated with collagenase, and then filtered through a 100 µm cell strainer (#431752CLS, Corning, NY, USA). The procured cells were embedded with growth factor-reduced Matrigel^®^ (#356231, Corning, NY, USA) on 24-well plates and cultured in a lung organoid growth medium of advanced DMEM/F-12 (#12634010, Gibco, Waltham, MA, USA) with GlutaMAX (#35050061, Gibco, Waltham, MA, USA), B-27 (#17504044, Gibco, Waltham, MA, USA), antibiotic-antimycotic (#15240096, Gibco, Waltham, MA, USA), 10 mM HEPES (#15630080, Gibco, Waltham, MA, USA), 10 mM nicotinamide (#N0636, Sigma-Aldrich, St. Louis, MO, USA), 1 mM N-acetylcysteine (#A9165, Sigma-Aldrich, St. Louis, MO, USA), 50 ng/mL hEGF (#E9644, Sigma-Aldrich, St. Louis, MO, USA), 500 nM A83-01 (#2939, Tocris Biosciences, Bristol, UK), 50 µg/mL Primocin (#ant-pm-1, InvivoGen, San Diego, CA, USA), 114 ng/mL hFGF7 (#100-19, Peprotech, Cranbury, NJ, USA), 112 ng/mL hFGF10 (#100-26, Peprotech, Cranbury, NJ, USA), a Wnt3A-conditioned medium (Gradiant Bioconvergence, Seoul, Republic of Korea), 2.5 µg/mL RSPO1 (Gradiant Bioconvergence, Seoul, Republic of Korea), and 100 ng/mL Noggin (Gradiant Bioconvergence, Seoul, Republic of Korea). During the first three days, the organoids were cultured with a lung organoid growth medium including a 10 µM ROCK inhibitor and Y-27632 (#ALX-270-333, Enzo Life Sciences, Farmingdale, NY, USA), after which the medium was replaced with a lung organoid growth medium without Y-27632. The medium was changed every three days. The organoids were sub-cultured every two weeks with TrypLE Express (#12604013, Gibco, Waltham, MA, USA) at a ratio of 1:2–1:3. 

### 2.2. Organoid-Derived Transitional Differentiated Cell Culture

Originating from a normal lung organoid between passages 5 and 7, Matrigel-embedded transitionally differentiated (TD) cells were harvested using trypsin/EDTA (#CC-5012, Lonza, Basel, Switzerland) and a trypsin-neutralizing solution (TNS) (#CC5002, Lonza, Basel, Switzerland), and organoid-derived TD cells were seeded at a density of 2 × 10^5^ cells per well in a 6-well plate. The cells were cultured in three different cell culture media: a lung organoid growth medium and a primary human airway epithelial cell expansion medium (PneumaCult EX-plus medium) (#05040, STEMCELL technologies, Vancouver, BC, Canada). For the ALI culture, the cells were seeded at a density of 5 × 10^4^ cells per well in a 24-well transwell plate coated with 2% growth factor-reduced Matrigel and cultured in a PneumaCult Ex-plus medium until full confluency was reached, after which the old medium was removed from the apical chambers and the bottom chamber medium was exchanged for a PneumaCult-ALI maintenance medium (#05001, STEMCELL technologies, Vancouver, BC, Canada). Seven days after forming the ALI culture, the apical chamber was washed once a week to remove mucous, while the cells were maintained under ALI culture conditions for seven weeks. 

### 2.3. Sample Preparation for scRNA-Seq

For scRNA-seq, normal lung tissues were acquired from two patients at the time of their lobectomy. Fresh tissues were isolated using a series of optimized lung tissue dissociation procedures and cultured according to the assumed cell differentiation types. For scRNA-seq analysis, we employed four types of cell suspensions, both organoid and organoid-derived TD cells, cultured in a lung organoid growth medium, a PneumaCult EX-plus medium, and a PneumaCult-ALI maintenance medium. Single-cell suspensions were prepared from the lung cells for the scRNA-seq library. We performed a general cell harvesting procedure on cell types of either organoid or organoid-derived TD cells, and the cell suspension was obtained through sequence steps corresponding to the cell preparation guide (CG00053 Rev C, 10× Genomics, Pleasanton, CA, USA) and adjusted to up to 5000 target cells with >95% viability. Using the Chromium Next GEM Single Cell 3-Gen Expression v3.1 Dual Index (CG00315 Rev D, 10× Genomics, Pleasanton, CA, USA) and a Chromium Single Cell 3’ Reagent Kit (PN1000268, 10× Genomics, Pleasanton, CA, USA), we generated GEM from those cell suspensions. Subsequently, using continuous procedures, including cDNA amplification and clean-up, we produced scRNA-seq libraries. Quality control of the scRNA-seq libraries was performed using an Agilent Bioanalyzer 4150 (Agilent Technologies, Santa Clara, CA, USA). The scRNA-seq libraries were analyzed on the Illumina NovaSeq 6000 platform (Illumina, San Diego, CA, USA) using paired-end reads and over 20,000 reads per cell.

### 2.4. scRNA-Seq Quality Control, Normalization, and Data Integration

Count matrices were generated using Cell Ranger (v7.0.1), by aligning the sequenced reads to the GRCh38 human reference genome. Each sample was subjected to ambient RNA removal using SoupX (v1.6.2), doublet removal using Doublet Finder (v2.0.3), and quality control filtering at gene and cell levels. To filter the data without removing biologically relevant cell types, outlier cells were determined based on the median absolute deviation (MAD) using Scater (v.1.0.4). This statistical filter was used for unique molecular identifier (UMI) counts, gene quantity, the mitochondrial count ratio, and the novelty score of each cell. The novelty score, calculated as the ratio of the gene counts to the UMI counts, was used to evaluate the complexity of RNA species in filtering out artifacts or contamination. Low-quality cells and genes were excluded, including genes expressed in fewer than 10 cells, cells with a total UMI count outside the range of three MAD from the median, cells with a total gene count outside the range of three MAD from the median, cells with a mitochondrial count ratio higher than three MAD from the median, and cells with a novelty score lower than three MAD from the median.

Only the cells that passed the doublet and MAD outlier filters were used for further analyses. SCTransform (v2) in the Seurat package (v4.3.0) was used for normalization and the percentages of mitochondrial genes and cell cycle scores were regressed. Canonical correlation analysis (CCA) and the Seurat standard analysis pipeline were used to conduct an integrative study of the eight samples constituting the atlas. The cell types were identified by annotation using scHCL (v0.1.1) [12], Azimuth (v0.4.6) [13], OSCA [14], and manual curation with canonical marker gene expressions. 

### 2.5. RNA Velocity and Pseudotime Analyses

The RNA velocity was estimated in the O1 (3D organoid) and ALIEX (organoid-derived TD cells with the PneumaCult-ALI maintenance medium) models to elucidate the differences in velocity pseudotimes. The pseudotimes of the two models were computed by integrating the RNA velocity information with gene expression using scVelo (v0.2.4).

### 2.6. Tissue Similarity Evaluation

Evaluation of the similarity with in vivo tissues was performed using two analysis algorithms. The WSAS lung tissue similarity scores were calculated using the Web-based Similarity Analysis System (W-SAS, v1) [15] algorithm with TPM-normalized pseudobulk gene expression data for each culture model. Other similarity scores were obtained from the prediction scores of the label-transfer algorithm using the Seurat package. This algorithm calculates the similarity in prediction scores between the tissue reference dataset and the sample dataset, using the identified anchors between the datasets to create a binary classification matrix that contains the classification information for each anchor cell in the tissue reference dataset. The prediction score for each class of each cell in the sample dataset was calculated using the binary classification matrix and the transpose of the weight matrix. Tissue reference datasets were obtained from the Integrated Human Lung Cell Atlas (HLCA) dataset [13]. We generated each tissue reference dataset by downsampling all classifications with the same number of cells using only normal datasets.

### 2.7. Differential Gene Expression Analyses and Further In-Silico Analyses Using iPathwayGuide

Differentially expressed gene (DEG) analysis comparing the O1 and ALIEX models (basal and secretory cells) was performed using DESeq2 (v1.30.1) after creating pseudobulk samples for each cell type in each model. The *p*-value was adjusted using the Benjamini–Hochberg method. DEGs were further analyzed using the Impact Analysis Method (IAM) [16,17] in the iPathwayGuide software (v17.0, Advaita Bioinformatics, Ann Arbor, MI, USA). These in silico analyses included signaling pathways, gene ontologies, gene regulatory networks, predictions of gene activation states, and alternative chemicals based on the Advaita Knowledge Base (AKB v17.0) [18]. The Kyoto Encyclopedia of Genes and Genomes (KEGG) database (Release 100.0+/11-12, Nov 21) [19,20,21] was used to decipher differentially regulated pathways, the Gene Ontology Consortium database (2021-Nov4) [22,23,24] was used to identify the differentially regulated gene ontology (GO) functions, the Biological General Repository for Interaction Datasets (BioGRID, v4.4.203) [25] and the STRING Database (v11.0) [26] were used to construct gene regulatory networks (GRNs), and the Comparative Toxicogenomics Database (Nov 2021) was used to find upstream chemicals. Significantly affected DEGs, pathways, and GO terms were analyzed using the IAM and compared with the corresponding control group. The *p*-values were computed based on a hypergeometric distribution. Fisher’s standard method was used to combine the *p*-values into one test statistic, and the *p*-value was adjusted based on the false discovery rate (FDR).

## 3. Results

### 3.1. Four Distinct Culture Methods on Adult Tissue-Derived Human Lung Organoids

Human lung organoids were generated from the fresh normal lung tissue of two individuals (R1 and R2). Surgically resected normal human lung tissue was enzymatically dissociated into single cells and used to generate lung organoids (O1 model). Lung organoid-derived TD cells were cultured in a lung organoid growth medium (O2 model), a PneumaCult EX-plus medium (EX model), or a PneumaCult-ALI maintenance medium (ALIEX model). The O1, O2, EX, and ALIEX models were generated, with each replicated using normal lung tissue from two individuals, totaling eight samples. Immunofluorescence (IF) staining with several epithelial cell markers was performed to confirm differentiation of the lung organoids into the epithelial lineage (Figure 1). Notably, an alveolar marker, HT I-56, was confirmed in the O1 3D organoid models, while a cilia marker, β-IV tubulin, was identified in the ALIEX models. 

### 3.2. Different Cell Composition of Each Culture Model Identified by scRNA-Seq

A single-cell transcriptional atlas of the eight samples was generated. A 10 × genomics platform was used to measure the single-cell transcriptomes. A total of 23,661 cells were detected; the median number of genes per cell exceeded 5500, and the alignment rate to the sample genome exceeded 95% in all samples. A total of 15,806 cells passed the quality control, and we obtained 2263 cells from the O1_R1 sample, 3331 from the O1_R2 sample, 2015 from the O2_R1 sample, 1571 from the O2_R2 sample, 1334 from the EX_R1 sample, 1857 from the EX_R2 sample, 1302 from the ALIEX_R1 sample, and 2133 from the ALIEX_R2 sample. Appendix A shows the UMI counts, gene quantities, mitochondrial count ratios, and novelty scores of each cell that passed the quality filters. Characteristically, the ALIEX model displayed a higher mitochondrial count ratio than the other models. Each dataset was integrated using CCA batch correction.

The major airway epithelial cell types (Figure 2A) were annotated using canonical cell-type marker genes (Figure 2C and Appendix A). The results showed that all four models contained basal, secretory, and ciliated cells, despite variations in their proportions (Figure 2D). Cell identity was inferred from the expression of specific marker genes, such as *KRT5* and *TP63* in basal cells, *SCGB1A1* and *MUC5AC* in secretory cells, and *FOXJ1* in ciliated cells. The annotated cell types were validated by comparison with three public datasets (HLCA, HCL, and OSCA) (Appendix A). The cell composition changed under ALI conditions, with a significant reduction in basal cells and an increase in ciliated cells (Figure 2D).

### 3.3. Identification of Basal Cell Subtypes

Basal cell clusters identified five additional subpopulations: proliferating basal, basal, differentiating basal, suprabasal, and transitioning basal (Figure 2B). Given the active differentiation of the organoids, we were able to capture the cellular transitional states that were readily apparent in homeostatic lungs. For example, the differentiation of the basal and suprabasal cells was captured. Differentiating basal cells expressed differentiation marker genes (*SERPINB3* and *NOTCH3*) and basal cell marker genes (*KRT5* and *TP63*), whereas suprabasal cells expressed reduced levels of typical basal cell marker genes (*KRT5* and *TP63*) but co-expressed suprabasal (*KRT19*) and secretory cell marker genes (*SCGB1A1*), suggesting the process of changing from basal to secretory cell states. In addition, a subset of basal cells actively expressing the proliferation marker gene (*MKI67*) was identified and annotated as proliferating basal cells. Further, one of the identified atypical subpopulations of basal cells was termed “transitioning basal”, because their gene expression profiles did not allow for precise classification with a reduced expression of typical basal cell marker genes and an increased expression of the transitioning alveolar basal cell marker gene (*KRT8*). These “transitioning basal” cells that we detected in our single-cell profiling resemble *CXCL14-* and *LGR6*-expressing (Figure 2C and Appendix A) TAB (transitioning airway basal) cells recently found in human lung airway organoids [14]. Transitioning basal cells decreased with the change from the 3D lung organoid model (O1) to the other 2D TD epithelial models (O2, EX, and ALIEX) (Appendix A).

### 3.4. The ALIEX Model Most Closely Resembles the In Vivo Human Lung Parenchymal Airway Epithelium

To evaluate which model better recapitulated the human lung tissue, we comparatively analyzed the transcriptomes and the cell population composition. The transcriptomic similarity was evaluated at two levels: tissue type (Figure 3A) and cell type (Figure 3B). The ALIEX model showed the highest transcriptomic similarity with W-SAS lung tissue and HLCA normal lung parenchymal airway epithelial tissue (Figure 3A). The EX model was the most similar to the HLCA normal airway epithelium. Notably, the ALIEX model showed a higher similarity score than the O1 model for all three tissues.

Because our target tissue to recapitulate is the normal lung airway epithelium, we next evaluated the similarities of each model with the HLCA normal lung parenchymal tissue datasets annotated as the airway epithelium (Figure 3B,C). The transcriptomic tissue similarity in the ciliated cell types was identical in all models (Figure 3B). In contrast, the basal and secretory cell types showed differences in tissue similarity between the models. The models most similar to tissue were the EX model in the basal cell type and the ALIEX model in the secretory cell type. The O1 model was the least similar to tissue in both the basal and secretory cell types. 

The cell compositions of normal lung parenchymal tissue, annotated as the airway epithelium, were characterized by containing the highest proportion of ciliated cells (60% on average), an average of 36% secretory cells, and the lowest proportion of basal cells (4% on average) (Figure 3C). Conversely, most culture models exhibited the highest proportion of basal cells and the lowest proportion of ciliated cells. This implies that the differentiation of ciliated cells from basal cells is insufficient in most culture models. Notably, a difference was observed in the ALIEX model: the proportion of ciliated cells was 12.5%, which was significantly higher than the average proportion in the other culture models (3.3%), and the proportion of basal cells was significantly lower (approximately half). Although this is still much lower than the high proportion (60% on average) shown in the tissue datasets, it is of great significance that ciliated cell differentiation significantly increased when the ALI culture method was used.

### 3.5. Differential Transcriptomic Signatures Associated with Epithelial Cell Alterations in the ALIEX Model Versus the O1 Model

To identify the key regulators of elevated ciliated cell differentiation in the ALIEX model, we directly compared the ALIEX and O1 models. 

First, a significant change in cell composition was observed between the ALIEX and the O1 models (Figure 4A and Appendix A). The ALIEX model is characterized by a predominance of three major cell types typically found in tissues (basal, secretory, and ciliated) [27,28]. In contrast, the O1 model contained over 30% of atypical basal subtypes (differentiating basal, suprabasal, and transitioning basal cells), whereas the ALIEX model accounted for less than 10% of these atypical subtypes. Notably, transitioning basal cells were present in the O1 model, whereas ciliated cells were abundant in the ALIEX model. In both models, the basal and secretory cells were the two cell types with composition proportions exceeding 10%.

Next, the pseudotime in each cell type was measured and compared between the ALIEX and O1 models (Figure 4B). Comparing the pseudotimes of the basal and secretory cells, cell types that are sufficiently abundant in both models, a significant difference was observed between the two models (Appendix A). The O1 model showed progression from basal to secretory cells, whereas the ALIEX model exhibited less distinct pseudo-time values between the basal and secretory cells. Secretory cells are known to dedifferentiate into basal cells in response to various stimuli [29,30,31,32]. Similarly, different culture conditions may influence cell composition (Appendix A) through dedifferentiation.

A closer analysis of the transcriptional signatures between the ALIEX and O1 models was conducted for the basal and secretory cells (Figure 4C). As a result, 2063 significant (*adjp* < 0.01) DEGs were identified in basal cells (Appendix A) and 4137 in secretory cells (Appendix A). In both cell types, the ALIEX model exhibited a greater number of genes with significantly increased expression than those with decreased expression. The number of DEGs with increased expression common to both cell types was 750, whereas those with decreased expression common to both cell types numbered 370.

Gene Ontology (GO) analyses were conducted separately for the DEGs from basal and secretory cells, followed by a meta-analysis comparing the outcomes of these two analyses (Figure 4D). The *p*-value was computed using the IAM hypergeometric distribution and corrected for multiple comparisons using FDR. According to the GO meta-analysis, 648 biological process GO terms were significant (*p_fdr* < 0.05) in basal cells (Appendix A) and 1003 were significant in secretory cells (Appendix A), with 429 significant biological process GO terms shared between the two cell types. By examining the GO terms related to epithelial cell alteration, it was confirmed that the ALI method induced basal-specific changes in epithelial cell migration and mesenchymal transition, as well as secretory-specific changes in epithelial cell proliferation and tube formation. Notably, epithelial cell differentiation was found to be the most significant (*p_fdr* 6.566 × 10^−6^ for basal cell and 2.545 × 10^−7^ for secretory cell) biological process GO term associated with epithelial cell alteration for both cell types. These findings are consistent with the known characteristic that both basal and secretory cells are undifferentiated stem-like cells capable of differentiating into more specialized cell types, including ciliated cells [33,34,35].

Therefore, pathway analysis was conducted for epithelial cell differentiation (GO: 0030855), a GO term with high significance in both populations. Significant DEGs (*adjp* < 0.05) associated with the GO term 0030855 (epithelial cell differentiation) were identified and a GRN (Figure 4E) was constructed using only DEGs with a matched direction of Log2FC in both basal and secretory cells. Among the interactions of this GRN, those with the highest confidence scores (900) given by the STRING database [26] were *HIF1A*-*VEGFA*, *CREB1*-*VEGFA*, and *KDM5B*-*CDKN1A* (Figure 4F). The significance of each gene in this GRN was assessed by scoring the centrality and closeness (Figure 4G) in each of the two populations, with *VEGFA* and *CDKN1A* identified as the most crucial. 

Gene activity prediction was conducted using the downstream genes for each of the nine genes (*VEGFA*, *CDKN1A*, *HIF1A*, *BMP4*, *CXCR4*, *CASP3*, *KDM5B*, *EPHA2*, and *CREB1*) with coherent edges (Figure 4E). *KDM5B*, *CXCR4*, *HIF1A*, and *VEGFA* showed consistent coherence between the predicted inhibited state and the measured expression changes (Appendix A). However, *CREB1* was predicted to be activated despite its downregulated expression (Appendix A). *CASP3*, *BMP4*, *CDKN1A*, and *EPHA2* were not predicted because of an insufficient number of DEGs corresponding to their downstream genes. To avoid confusion due to expression changes that are inconsistent with the actual activation state, we excluded *CREB1*, which has a high risk of discrepancy, and set the other eight genes as targets.

### 3.6. In-Silico Chemical Screening for Promoting Ciliated Cell Differentiation

To identify the key signaling pathways, an extended GRN (Figure 5A) was constructed by incorporating genes from the KEGG pathways, centering on eight target genes (*VEGFA*, *CDKN1A*, *HIF1A*, *BMP4*, *CXCR4*, *EPHA2*, *CASP3*, and *KDM5B*) with significant expression changes induced by the ALI culture method in both basal and secretory cells. Two major pathways were identified in this extended GRN: a downregulated pathway involving *HIF1A*/*BMP4*-*VEGFA* and an upregulated pathway leading to *STAT3*-*TP53*-*CDKN1A* (Appendix A). Further, *STAT3*, *TP53*, *CREB1*, and *HIF1A*, which are known regulators of both *VEGFA* and *CDKN1A*, displayed divergent effects on these two genes within this network. Therefore, we aimed to identify the chemicals that regulate the expression of two central genes, *VEGFA* and *CDKN1A*, which are at the center of major streams and exhibit coherent downstream signals.

Specifically, after identifying 2082 genes (Appendix A) with significant (*adjp* < 0.05) and coherent changes in both basal and secretory cells, an upstream regulator analysis based on the DEG patterns and literature references was conducted. This comprehensive analysis included all 2082 genes to ensure that genes potentially involved in epithelial cell differentiation were not excluded, even if they were not part of the GO term 0030855 (epithelial cell differentiation). As a result, 64 significant upstream chemicals were identified (Appendix A).

We individually verified whether each of these 64 chemicals regulated eight target genes (*VEGFA*, *CDKN1A*, *HIF1A*, *BMP4*, *CXCR4*, *EPHA2*, *CASP3*, and *KDM5B*), particularly *VEGFA* and *CDKN1A*, in the desired direction. Figure 5B summarizes the chemicals predicted to regulate the eight target genes in the desired direction. Notably, flavanones, fulvestrant, dexamethasone, and camptothecin were predicted to directly downregulate *VEGFA* while simultaneously upregulating *CDKN1A*. Figure 5C illustrates the actions of several chemicals that regulate the two major signals identified. These chemical candidates are expected to promote ciliated cell differentiation by mimicking the key signals of epithelial cell differentiation identified in the ALIEX model.

## 4. Discussion

Our demonstration of enhanced ciliated cell differentiation in ALI culture addresses key challenges in the transition to human-derived cells for preclinical testing, which is driven by ethical concerns surrounding animal testing and differences in immune system responses between humans and animals [36,37]. In vitro technologies such as organ-on-a-chip and organoids are leading this transition, aiming to create models that more accurately simulate human physiology and potentially replace traditional animal models in drug development and testing [38,39]. 

To enable in vitro alternative modeling, it is essential to identify the regulatory factors at each stage of organ development and cell differentiation. However, the mechanisms underlying airway epithelial development are highly complex and not yet fully understood [40]. Identifying the mechanisms and regulatory factors involved in the differentiation of ciliated cells, the most specialized cell types in the airway epithelium, is crucial for developing accurate in vitro airway epithelium models using human cells.

In this study, we analyzed the differentiation effects and regulatory factors of the ALI culture method on adult tissue-derived human lung organoids using scRNA-seq.

Our comparative analysis of four distinct culture models revealed significant differences in differentiation outcomes, particularly between the submerged and ALI methods (Figure 1). The O1 model represents a submerged organoid culture method, whereas the ALIEX model represents an air-exposed ALI culture method. Two submerged TD models were generated to represent the intermediate states between the O1 and ALIEX models. Cell composition and tissue similarity analyses were conducted for the four models (Figure 2D, Figure 3C and Appendix A). First, cell composition analysis (Figure 2D) revealed a high proportion of ciliated cells and a low proportion of basal cells in the ALIEX model, which clearly distinguished it from the other three models. Second, tissue similarity analysis (Figure 3A) using transcriptomic profiles of the pseudobulk for each model revealed that the ALIEX model exhibited the highest lung similarity compared to the other models, confirming the promise of the ALI culture method for in vitro lung modeling. When a finer level of annotation of lung tissue reference data (HLCA) was used, the ALIEX model showed the highest similarity to the lung parenchymal airway epithelium. However, when tissue similarities (HLCA lung parenchymal airway epithelium) were evaluated for each of the three major cell types (basal, secretory, and ciliated cell) (Figure 3B), the ALIEX model did not display particularly high tissue similarities. This suggests that the high tissue similarity of the ALIEX model is due to its overall cell composition (Figure 3C) rather than the transcriptomic profiles of the individual cell types (Figure 3B), indicating that generating differentiated cell populations at levels comparable to those in tissues is essential for achieving high tissue similarity. Although the ALIEX model had the most tissue-like cell composition of the four models, it differed from the actual tissue (HLCA lung parenchymal airway epithelium) (Figure 3C). The basal cell proportion in the tissue reference data was approximately 4%, and the proportion of ciliated cells was approximately 60%, compared to 25% and 13%, respectively, in the ALIEX model. 

Although recent studies [14,41,42] have reported ciliated cell production exceeding 10% using 3D lung organoids, the efficiency of ciliated cell production can be influenced by multiple factors, including donor conditions, culture conditions, the medium composition, and the temporal dynamics of differentiation [43,44,45]. In our experimental conditions, although we observed a relatively lower proportion of ciliated cells compared to these studies, we consistently found the highest proportion of ciliated cells in the ALIEX model among our tested models (Figure 2D). This observation aligns with previous studies, affirming that the ALI method effectively enhances ciliated cell differentiation [11,46]. Since the proportion of ciliated cells showed a significant improvement in the ALIEX model compared to the O1 model (Figure 4A), we conducted a comparative analysis to identify the differentially activated regulatory factors between these two models.

Our GRN analysis identified key regulatory factors driving the enhanced differentiation observed in ALI culture by comparing the DEGs between the ALIEX and O1 models (Figure 4E). From the DEGs present in both the basal and secretory cells (Figure 4C), we identified nine coherent common DEGs involved in epithelial cell differentiation (*VEGFA*, *CDKN1A*, *HIF1A*, *BMP4*, *CXCR4*, *EPHA2*, *CASP3*, *KDM5B*, and *CREB1*) (Figure 4E). After validating the correspondence between expression changes and activation states (Appendix A), we focused on eight genes (*VEGFA*, *CDKN1A*, *HIF1A*, *BMP4*, *CXCR4*, *EPHA2*, *CASP3*, and *KDM5B*), excluding *CREB1* due to inconsistent activation patterns (Figure 5C). The identification of these eight consistent regulators across both basal and secretory cells suggests a common molecular mechanism underlying ALI-induced differentiation, rather than cell-type specific responses. This finding is particularly significant as it indicates that targeting these shared pathways could potentially enhance differentiation efficiency without the need for cell-type specific interventions.

Our extended network analysis with KEGG pathway genes revealed two major signaling pathways in the ALIEX model: downregulation of *VEGFA*, a downstream gene of *HIF1A*, and upregulation of *CDKN1A* (Figure 5A). Because physically separating basal and secretory cells during differentiation from the same origin in vitro is challenging, we focused on identifying chemicals that can promote ciliated cell differentiation by targeting these regulatory genes shared by both cell populations. This approach minimizes the risk of intersection-related side effects. Using the AKB database, we identified several chemical candidates, including flavanones, fulvestrant, dexamethasone, and camptothecin, that could potentially alternate the ALI culture method by modulating these key regulatory pathways (Figure 5B,C). By leveraging these common regulatory signals, specifically the downregulation of *VEGFA* and the upregulation of *CDKN1A*, we expect these chemical candidates to drive ciliated cell differentiation more effectively and improve the accuracy of in vitro lung models. 

Although ALI culture is known to promote cell differentiation, including the production of ciliated cells [11,46], the mechanisms and regulatory factors involved in ciliated cell differentiation have not been sufficiently studied. Several studies have shown that oxygenation, the ALI culture method, and the deactivation of *HIF1A* are associated with ciliated cell production [47,48,49,50,51]. However, these studies validated these findings at the protein level. To the best of our knowledge, this is the first study to examine the effect of ALI on ciliated cell differentiation. We identified two key regulatory factors (*HIF1A*-*VEGFA* and *CDKN1A*) involved in ciliated cell differentiation. These findings elucidate the critical regulatory pathways by which ALI promotes the differentiation of ciliated cells and provide valuable insights for refining in vitro human lung airway models.

The current study had some limitations. First, this study compared the results of four distinct culture methods but did not track changes over time within the ALI culture method. Sampling the ALI model at several time points and conducting comparative analyses will enable a more in-depth understanding of the proportions of ciliated cells, changes in gene expression, and key regulatory factors over time. Additionally, the study included only two donors and lacked technical replicates for the scRNA-seq experiments. This limited sample size imposes constraints on our findings, highlighting the need for further research to validate and expand upon these results.

Second, several recent studies have reported ciliated cell production of over 10% using 3D lung organoids. To identify the regulatory signals that directly influence ciliated cell differentiation, comparing and analyzing samples with varying differentiation rates, even when using the same culture method, may prove more effective.

Third, verification experiments on the identified key regulatory genes are essential. Previous studies have demonstrated that reducing *HIF1A* expression through oxygenation or *HIF1A* inhibitor treatment promotes ciliated cell differentiation [52,53]. However, the role of *VEGFA* inhibition in the induction of ciliated cell differentiation remains unclear. *VEGFA*, a crucial factor in progenitor cell differentiation into alveolar lineages and a marker gene of alveolar type I cells [54,55,56,57], needs to be tested to confirm whether its inhibition redirects differentiation towards airway lineages and promotes ciliated cell differentiation. In addition, the previously reported involvement of *CDKN1A* in ROS production and cell cycle arrest [58,59,60,61,62,63,64] needs to be explored to confirm whether the increased *CDKN1A* expression induced by specific oxygenation levels induces cell cycle arrest and promotes ciliated cell differentiation.

## 5. Conclusions

In conclusion, our study provides insights into the enhancement of epithelial cell differentiation for the development of an in vitro lung airway model using human lung organoids, highlighting two key regulatory pathways, *HIF1A*-*VEGFA* and *CDKN1A*, as the drivers of epithelial cell differentiation.

The ALI method, which promotes epithelial differentiation by exposing cells to air, resulted in a more than fivefold increase in ciliated cell production compared to the submerged culture methods. Despite the effectiveness of the ALI method, ciliated cell proportions are still less than one fourth of those found in human lung airway tissues. Through comparative analysis, we identified these regulatory pathways as being central to ciliated cell differentiation and proposed several chemical candidates that regulate these signaling pathways. Future investigations should focus on time-series comparative analyses within the ALI model and validate the impact of the identified key regulators and chemical candidates on ciliated cell differentiation.

## Figures and Tables

**Figure 1 cells-13-01991-f001:**
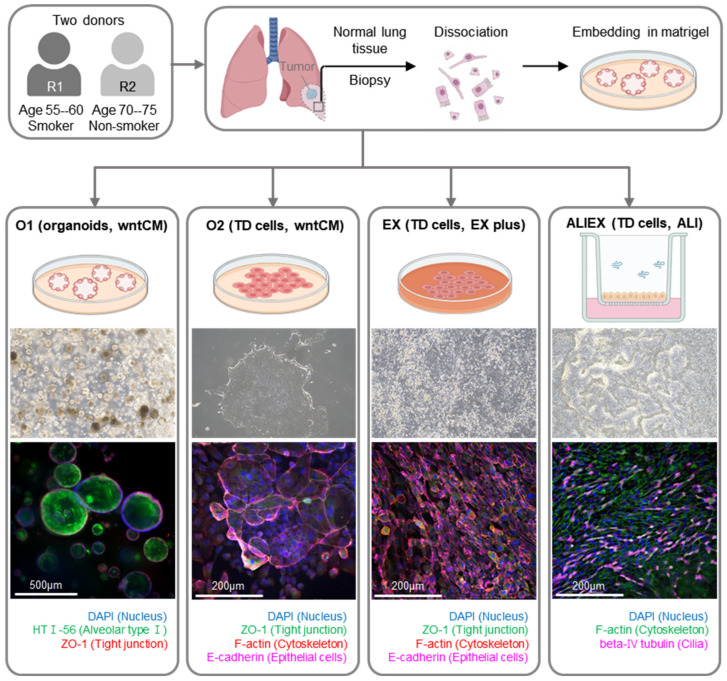
Schematic workflow and representative immunofluorescent images of four distinct culture methods on adult tissue-derived human lung organoids: O1 (3D organoid), O2 (2D culture with organoid wntCM media), EX (2D culture with PneumaCult EX plus media), and ALIEX (air–liquid interface culture with PneumaCult ALI media). Immunofluorescence was used to detect proteins associated with an alveolar, epithelial, or ciliated expression. The images are representative of four distinct culture experiments. Scale bars: 200 or 500 μm.

**Figure 2 cells-13-01991-f002:**
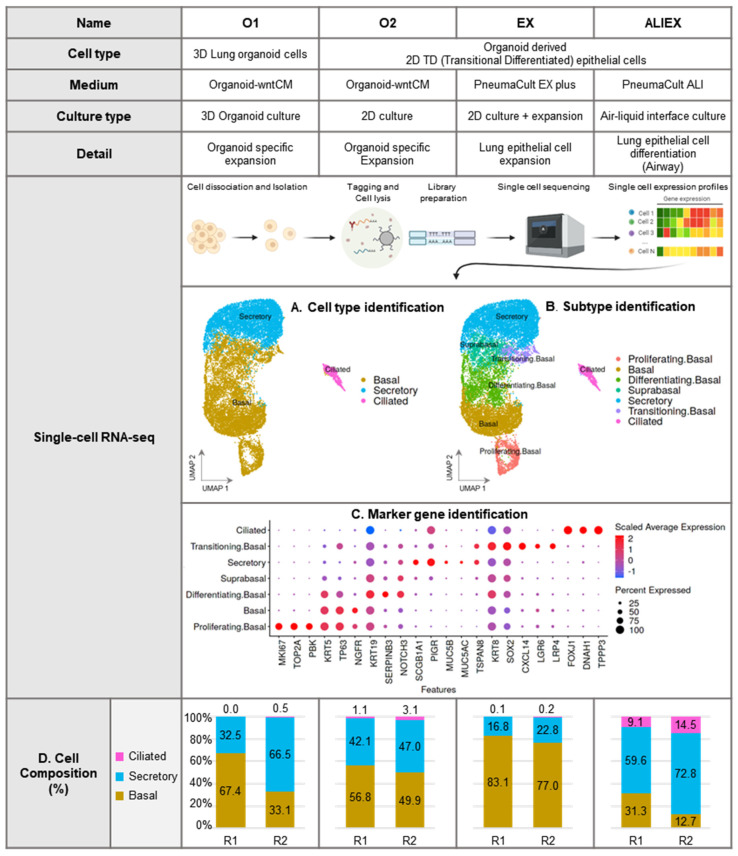
Differences in cell composition for each culture model of adult tissue-derived human lung organoids identified by single-cell RNA sequencing (scRNA-seq) analysis. (**A**,**B**) Uniform manifold approximation and projection (UMAP): visualization of the whole dataset. Each cell type is defined by a specific color. (**A**) Three major cell types (basal, secretory, and ciliated cells) were identified using canonical marker genes. (**B**) Five subtypes of basal cells (proliferating basal, basal, differentiating basal, suprabasal, and transitioning basal) were identified using known and conserved marker genes. (**C**) Dot plot of scaled and averaged expression values for the top marker genes of each cell type in the whole data set. (**D**) Bar graph of the major cell types identified for each biological replicate (R1 and R2) in each model.

**Figure 3 cells-13-01991-f003:**
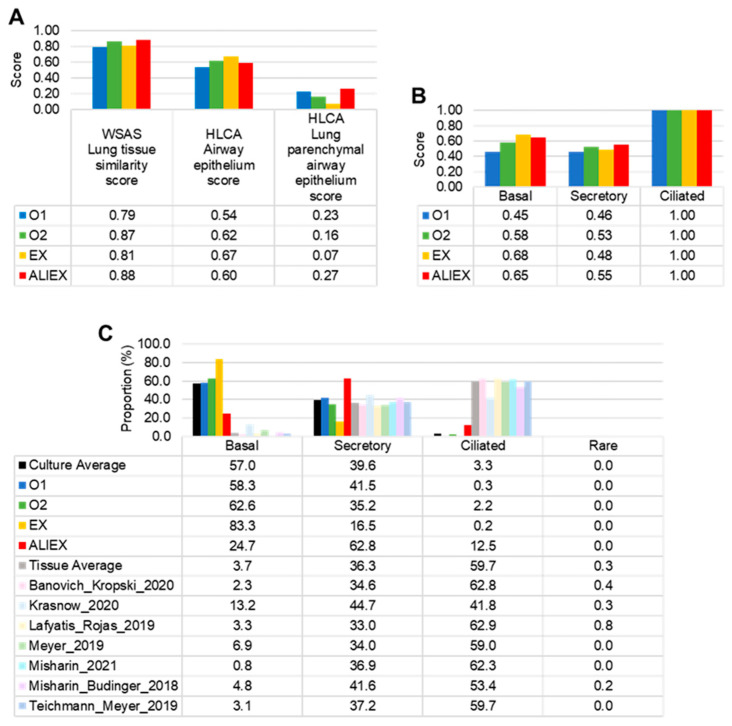
The ALIEX model closely resembles the in vivo human lung parenchymal airway epithelium. (**A**) Bar graph of the tissue similarity scores in each culture model for three different tissue datasets. The scores were calculated using the Web-based Similarity Analysis System (W-SAS) API or the Seurat package. (**B**) Bar graph of the cell type similarity scores for the three major cell types (basal, secretory, and ciliated) for each model of the Integrated Human Lung Cell Atlas (HLCA) lung parenchymal airway epithelium dataset. (**C**) Bar graph showing the cell type proportion (%) in the culture models and tissue datasets. The seven tissue datasets were separated from the HLCA lung parenchymal airway epithelium dataset.

**Figure 4 cells-13-01991-f004:**
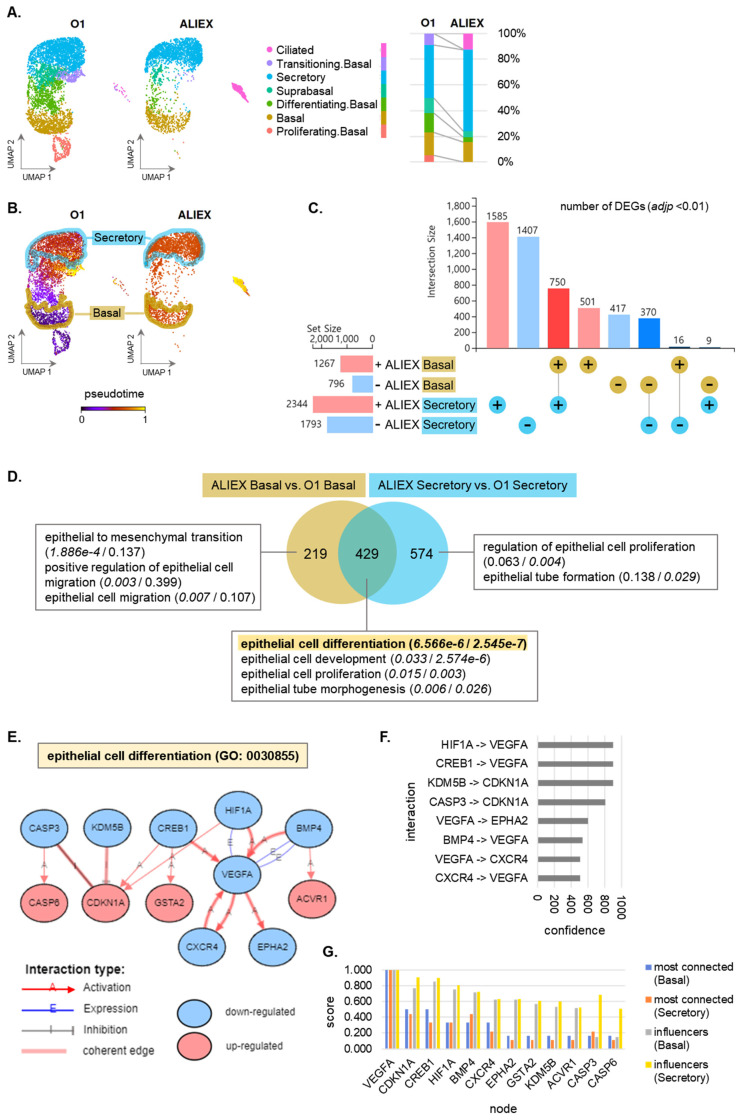
Transcriptomic signatures associated with epithelial cell alterations in the ALIEX model versus those in the O1 model. (**A**) UMAP visualization and composition bar graphs of each cell type in the O1 and ALIEX models. (**B**) Pseudotime picturing using a purple–red–yellow gradient along the differentiation trajectory in each model. The basal and secretory cells are highlighted. (**C**) Differential expression (DE) results of the differentially expressed gene (DEG) counts in ALIEX basal vs. O1 basal cells and ALIEX secretory vs. O1 secretory cells. Genes with a positive Log2FC (pink): +, genes with a negative Log2FC (sky blue): −. (**D**) A Venn diagram showing overlaps in the Gene Set (GO) terms between the DE results. The *p_fdr* of each GO term is in the blank (left: *p_fdr* in ALIEX basal vs. O1 basal DE analysis, right: *p_fdr* in ALIEX secretory vs. O1 secretory DE analysis). (**E**) The gene regulatory network (GRN) in GO 0030855 (epithelial cell differentiation) for DE results of shared DEGs in ALIEX basal vs. O1 basal and ALIEX secretory vs. O1 secretory. (**F**) Confidence score of each interaction in the network (**E**). (**G**) Bar graph of centrality (most connected) and closeness (influencers) scores in each node of the network (**E**).

**Figure 5 cells-13-01991-f005:**
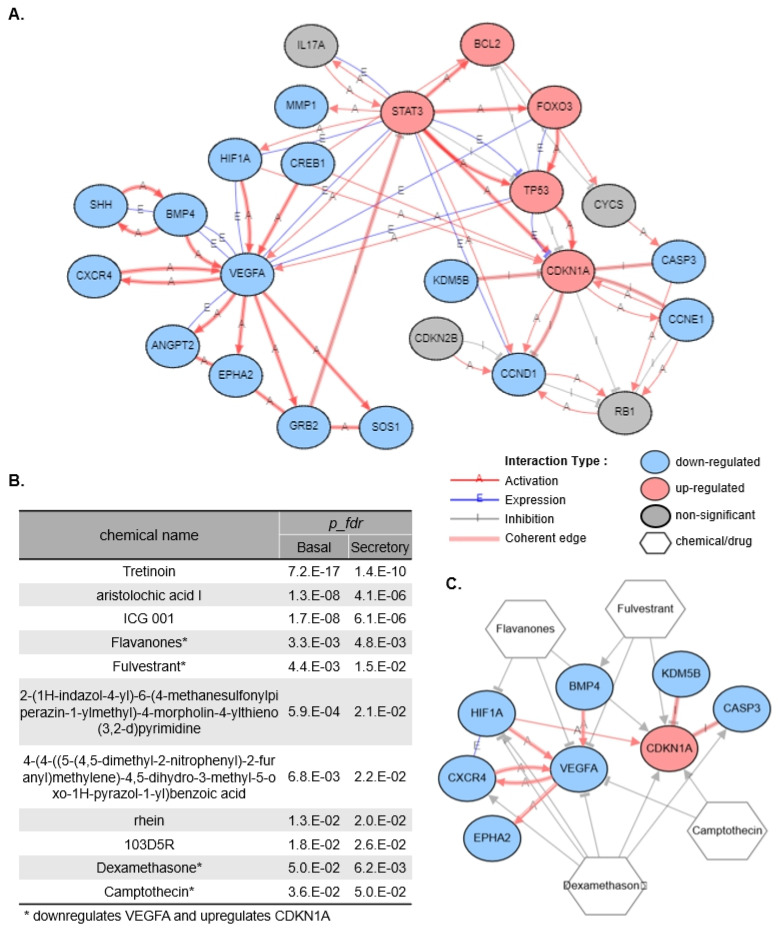
In-silico chemical screening of *VEGFA* inhibitors with *CDKN1A* activation properties. (**A**) The GRN network in Figure 4E was extended by adding genes from the KEGG pathway. (**B**) Upstream chemicals predicted as present in ALIEX basal vs. O1 basal and ALIEX secretory vs. O1 secretory DE results that regulate any of the 8 target genes (*VEGFA*, *CDKN1A*, *HIF1A*, *BMP4*, *CXCR4*, *EPHA2*, *CASP3*, and *KDM5B*). * indicates a chemical that downregulates *VEGFA* and upregulates *CDKN1A*. (**C**) GRN graph summarizing the regulatory effects of selected chemicals on the target genes.

## Data Availability

The datasets generated during the current study are available in the GEO repository with the GEO accession number GSE280502 (https://www.ncbi.nlm.nih.gov/geo/query/acc.cgi?acc=GSE280502, accessed on 16 November 2024).

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
