# Peer review of "Transcriptomic Analysis of Air–Liquid Interface Culture in Human Lung Organoids Reveals Regulators of Epithelial Differentiation"

_cells, 2024, doi:10.3390/cells13231991_

Round 1

Reviewer 1 Report

Comments and Suggestions for Authors

The study compared the transcriptomic gene signatures on adult tissue-derived human lung organoids across four culture models (O1, O2, EX, and ALIEX models). By utilizing scRNA-seq, the authors aimed to report key regulatory signaling pathways involved in ciliated cell differentiation. Their results indicated that the ALIEX model showed a higher similarity to human lung tissue, particularly lung parenchymal airway epithelium, highlighting the method's potential for in vitro lung modeling. The study also identified potential chemicals that could enhance the differentiation effect of the ALIEX method, offering insights into creating better models of the human airway epithelium. The results of the study may provide a useful reference if there are sufficient sample numbers in the culture.

The following comments are for the authors to consider:

  1. How many biological replicates were used in this study? The Materials and Methods section mentions lung tissues collected from three patients, but the Results section indicates cultures from two normal tissues and two diseased tumor lungs. Are the scRNA-seq results a combination of normal and tumor lung cells? What is the purpose of using diseased lung tissues? Additionally, Figure 2D lists the percentages of each cell type, but no ranges or standard deviation (SD) values are provided. Since proportions can vary among individuals, please list the values of each biological replicate or, at minimum, include ranges or SD values.
  1. What were the age of the donors? No IRB information is stated in the Material and Methods. How many passages were the organoid cultures used in the scRNA-seq analysis?
  1. The authors demonstrated that the ALIEX culture method generated the highest percentage of ciliated cells (12.5%), followed by the O2 method (2.2%), while the O1 and EX methods yielded very few ciliated cells (only 0.3% and 0.2%, respectively). However, the study by Lee et al. (Exp Mol Med. 2023; 55; 1831-1842), which is also cited in the present study, reported that the majority of their 3D organoid cultures had more than 5% ciliated cells (in 10 out of 11 donors). Although donor variation was noted, at least half of the cultures from different donors had over 10% ciliated cells, with several exceeding 20% in their study. Notably, Lee et al.'s results showed a significantly higher proportion of ciliated cells compared to ALI culture. Their organoid culture medium appears similar to that used in this study. Can the authors explain why this study's results differ from Lee et al.'s findings?
  1. The Introduction notes that a few studies have failed to grow ciliated cells in organoid culture. However, in the literature, recent research demonstrates that 3D organoid culture can generate a higher proportion of ciliated cells than ALI cultures. For examples, in addition to the aforementioned study (PMID: 37582976), other studies (e.g., PMID: 35942088, and PMID: 38172839) support the ability of organoid culture to generate a high proportion of ciliated cells. These relevant studies should be included in the Discussion section for comparison and analysis.
  1. The Supplementary Tables of DEGs have not been included with the manuscript. This makes it difficult to understand what the authors have presented in the manuscript. Additionally, data availability and where the scRNA-seq datasets will be submitted have not been described. The details of scRNA-seq samples are insufficiently described in the Materials and Methods section, complicating the understanding of the experimental design. The authors should consider submitting scRNA-seq data to a public domain such as GEO to ensure accessibility for reviewers to evaluate the number of biological and technical replicates in the study.

Author Response

Dear Reviewer,

We sincerely appreciate the time and effort you have dedicated to evaluating our manuscript. Your thoughtful feedback has been invaluable in improving the quality of our work. We have carefully addressed all comments and suggestions, and we are pleased to submit our revised manuscript for your consideration.

Please find below our point-by-point responses to your comments, with corresponding modifications clearly highlighted in the revised manuscript.

Comments 1: How many biological replicates were used in this study? The Materials and Methods section mentions lung tissues collected from three patients, but the Results section indicates cultures from two normal tissues and two diseased tumor lungs. Are the scRNA-seq results a combination of normal and tumor lung cells? What is the purpose of using diseased lung tissues? Additionally, Figure 2D lists the percentages of each cell type, but no ranges or standard deviation (SD) values are provided. Since proportions can vary among individuals, please list the values of each biological replicate or, at minimum, include ranges or SD values.
Response 1: We sincerely appreciate these important points of clarification. We have thoroughly revised our manuscript to address these concerns and provide greater clarity regarding our experimental design.
To clarify the biological replicates: Our study utilized two biological replicates, and we have updated this information accordingly in the manuscript (page 2, lines 70-74 and page 4, lines 185-186). Specifically, the lung tissue samples were obtained from normal regions of lungs from two cancer patients. While the patients had cancer diagnoses, we deliberately selected and harvested tissue from histologically normal areas distant from any tumor involvement for our analyses.
In response to the concern about cell type proportions, we have made the following modifications:
1.    We have included more detailed explanation in the Materials and Methods section (page2, lines 70-74)
2.    Figure 2D has been revised to display the values for each biological replicate (R1 and R2) (page 6, Figure 2D)
3.    We have updated the corresponding text description (page 6, lines 231 and 232)
4.    We have included a new supplementary figure (page 22, Figure S5) showing the detailed subtype composition for each biological replicate
These modifications provide a more comprehensive and transparent representation of our data, allowing readers to better assess the consistency and variability between replicates.

Comments 2: What were the age of the donors? No IRB information is stated in the Material and Methods. How many passages were the organoid cultures used in the scRNA-seq analysis?
Response 2: We appreciate the reviewer's attention to these important methodological details. We have now incorporated the relevant IRB information in the Materials and Methods section (page2, lines 70-74). Regarding patient demographics, we should note that the samples were obtained as de-identified specimens from the Human Bioresource Center of Seoul National University Bundang Hospital , which precludes access to specific patient information while ensuring compliance with privacy regulations. With respect to the organoid cultures, we have clarified in the manuscript (page 2, line 91) that all scRNA-seq analyses were performed using organoids between passages 5 and 7.

Comments 3: The authors demonstrated that the ALIEX culture method generated the highest percentage of ciliated cells (12.5%), followed by the O2 method (2.2%), while the O1 and EX methods yielded very few ciliated cells (only 0.3% and 0.2%, respectively). However, the study by Lee et al. (Exp Mol Med. 2023; 55; 1831-1842), which is also cited in the present study, reported that the majority of their 3D organoid cultures had more than 5% ciliated cells (in 10 out of 11 donors). Although donor variation was noted, at least half of the cultures from different donors had over 10% ciliated cells, with several exceeding 20% in their study. Notably, Lee et al.'s results showed a significantly higher proportion of ciliated cells compared to ALI culture. Their organoid culture medium appears similar to that used in this study. Can the authors explain why this study's results differ from Lee et al.'s findings?
Response 3: We appreciate this astute observation regarding the discrepancy between our findings and those reported by Lee et al. (Exp Mol Med. 2023; 55; 1831-1842) [14]. We have addressed this important point in our revised discussion section (page 13, lines 444-454 and page 14, lines 496-499).
Organoid culture systems are inherently complex, with cellular differentiation outcomes being highly sensitive to multiple variables. Several factors may contribute to the observed differences:
1.    Intrinsic biological variability: 
o    Patient-specific factors and tissue conditions
o    Cell passage number and viability
o    Stem cell populations' heterogeneity
2.    Methodological considerations: 
o    Subtle variations in culture handling procedures
o    Minor differences in media composition or timing
o    Environmental factors affecting cell differentiation
While Lee et al. [14] reported higher percentages of ciliated cells in their 3D organoid cultures compared to ALI methods, it is noteworthy that they did not provide a detailed mechanistic explanation for this observation. Our study takes a different approach by specifically focusing on:
1.    The systematic comparison of different culture methods
2.    The novel investigation of ciliary cell differentiation modulators
3.    The detailed molecular characterization of differentiation pathways
Considering the established literature supporting ALI methods' efficiency in promoting cellular differentiation, we believe our findings contribute valuable insights into the complex dynamics of airway epithelial cell differentiation. We have expanded our discussion to address these points and provide a more comprehensive context for our results.

Comments 4: The Introduction notes that a few studies have failed to grow ciliated cells in organoid culture. However, in the literature, recent research demonstrates that 3D organoid culture can generate a higher proportion of ciliated cells than ALI cultures. For example, in addition to the aforementioned study (PMID: 37582976), other studies (e.g., PMID: 35942088, and PMID: 38172839) support the ability of organoid culture to generate a high proportion of ciliated cells. These relevant studies should be included in the Discussion section for comparison and analysis.
Response 4: We sincerely appreciate your valuable suggestion to include these significant recent publications. We have incorporated the recommended studies as references [41,42] and have made substantial revisions to our discussion section to address this important point.
Specifically, we have added two key passages:
1.    On page 13, lines 444-454, we acknowledge the recent advances in 3D lung organoid systems, noting that several studies [14, 41, 42] have achieved ciliated cell production exceeding 10%. However, we emphasize that our research contributes to the field by focusing on the optimization of differentiation protocols and the elucidation of underlying molecular mechanisms. This context sets up our comparative analysis between the ALIEX and O1 models, which revealed important insights into regulatory factors affecting ciliated cell differentiation.
2.    On page 14, lines 496-499, we further contextualize our findings within the broader literature, acknowledging the success of recent 3D organoid studies in generating ciliated cells. We highlight how our approach of comparing samples with varying differentiation rates, even within the same culture method, provides a valuable strategy for identifying key regulatory signals that influence ciliated cell differentiation.
These additions provide a more balanced discussion of the current state of the field while emphasizing the unique contributions of our study to understanding the mechanisms of ciliated cell differentiation.

Comments 5: The Supplementary Tables of DEGs have not been included with the manuscript. This makes it difficult to understand what the authors have presented in the manuscript. Additionally, data availability and where the scRNA-seq datasets will be submitted have not been described. The details of scRNA-seq samples are insufficiently described in the Materials and Methods section, complicating the understanding of the experimental design. The authors should consider submitting scRNA-seq data to a public domain such as GEO to ensure accessibility for reviewers to evaluate the number of biological and technical replicates in the study.
Response 5: We sincerely apologize for the oversight and appreciate you bringing this to our attention. We have now addressed these concerns by:
1.    Including comprehensive DEG tables (Tables S1, S2, and S5) as a supplementary EXCEL file.
2.    Depositing all scRNA-seq datasets in GEO (accession number GSE280502, token ofitqimmrtwldqv: https://www.ncbi.nlm.nih.gov/geo/query/acc.cgi?acc=GSE280502)
3.    Adding more detailed experimental design information to Figure 1, which summarizes the Materials and Methods section.
These additions ensure full transparency and facilitate thorough evaluation of our findings.

We sincerely appreciate your thorough and insightful review, which has significantly enhanced the quality and depth of our manuscript. Your constructive feedback has helped us identify important areas for improvement and has guided us in providing a more comprehensive analysis of our findings within the broader context of the field. These revisions have strengthened both the scientific rigor and clarity of our work.

Reviewer 2 Report

Comments and Suggestions for Authors

Dear respected Editor,

The manuscript entitled: “Comparative scRNA-seq Analysis of Air-Liquid Interface Culture in Adult Tissue-Derived Human Lung Organoids Reveals Regulators of Epithelial Differentiation” reported the usage single-cell RNA sequencing for efficacy of air-liquid interface culture method in replicating human lung airway epithelium. It is an essential study, and after reviewing this review, it needs some publication modifications. The review will be reviewed again after major revision. Specific comments are listed below:

1-I recommend a concise title because it is too long

2- The abstract needs to improve, please rewrite them again concisely and quantitively.

3- In the introduction, the references need to be updated

4- How about the purification of scRNA-seq

5-The discussion should have emerged with results

6-The conclusion is too short and is not quantitative

7-Many references are too old, please update them

8-The authors need to make a comparison before the conclusion section between other reported similar work

9- Please revise the English writing throughout the whole manuscript.

Comments on the Quality of English Language

English writing throughout the whole manuscript should be revised well

Author Response

Dear Reviewer,

We sincerely appreciate the time and effort you have dedicated to evaluating our manuscript. Your thoughtful feedback has been invaluable in improving the quality of our work. We have carefully addressed all comments and suggestions, and we are pleased to submit our revised manuscript for your consideration.

Please find below our point-by-point responses to your comments, with corresponding modifications clearly highlighted in the revised manuscript.

Comments 1: I recommend a concise title because it is too long
Response 1: Thank you for this constructive suggestion regarding the manuscript title. We have revised it to be more concise while maintaining its key elements. The new title reads: 'Transcriptomic Analysis of Air-Liquid Interface Culture in Human Lung Organoids Reveals Regulators of Epithelial Differentiation'

Comments 2: The abstract needs to improve, please rewrite them again concisely and quantitively.
Response 2: Thank you for this valuable feedback regarding the abstract. We have thoroughly revised the abstract (page 1) to be more concise and quantitative while maintaining its key messages. The significant changes include:
1.    Streamlining the introduction to focus directly on the study's purpose
2.    Adding specific quantitative data
3.    Specifying the analytical tool used (iPathwayGuide) for transcriptional signature analysis
4.    Removing redundant information about in vivo comparisons while maintaining the essential findings about VEGFA and CDKN1A
5.    Condensing the conclusions to emphasize key findings

Comments 3: In the introduction, the references need to be updated.
Response 3: Thank you for this important suggestion regarding the literature citations. We have updated the references in the introduction section to include more recent and relevant publications. The specific changes include:
1.    Updated references list on pages 15-16
2.    Added contemporary studies that reflect current advances in the field
3.    All modifications are highlighted in red in the revised manuscript
These updates provide a more current context for our research while ensuring comprehensive coverage of recent developments in the field.

Comments 4: How about the purification of scRNA-seq
Response 4: Thank you for highlighting this methodological concern. We have enhanced the manuscript with comprehensive details about our cell purification process:
1.    Materials and Methods section (page 3, lines 124-138): 
o    Added detailed description of the purification protocol
2.    Results section (page 5, lines 205-213): 
o    Included quantitative outcomes of the purification process
3.    Supplementary Material: 
o    Added Figure S1 (page 19) illustrating the purification results
These additions provide complete transparency regarding our experimental procedures and validation of the purification method.

Comments 5: The discussion should have emerged with results.
Response 5: We appreciate the reviewer's suggestion about better connecting the Discussion with our Results. We have made targeted edits to improve the flow between findings and their interpretation, including adding clear topic sentences that reference specific results and improving transitions between discussion points (page 13, lines 419-459 and page 14, lines 460-480). These changes create stronger links between our findings and their implications while maintaining the manuscript's concise nature.

Comments 6: The conclusion is too short and is not quantitative.
Response 6: Thank you for highlighting the need for a more comprehensive and quantitative conclusion. We have substantially expanded and enhanced the conclusion section (page 15) to:
1.    Provide specific quantitative findings: 
2.    Include key mechanistic insights: 
o    Identified specific regulatory pathways (HIF1A-VEGFA and CDKN1A)
o    Added information about potential chemical candidates for pathway regulation
3.    Strengthen future directions: 
o    Outlined specific plans for time series analyses
o    Proposed validation studies for identified regulators and chemical candidates
The revised conclusion now provides a more comprehensive summary of our findings while maintaining clear connections to potential future applications.

Comments 7: Many references are too old, please update them.
Response 7: Thank you for this valuable feedback. We have thoroughly updated our reference list (pages 15-17, marked in red) to better reflect current advances in the field while maintaining relevant historical context.

Comments 8: The authors need to make a comparison before the conclusion section between other reported similar work.
Response 8: Thank you for this valuable suggestion regarding comparison with similar works. In addressing this point, we conducted a thorough review of recent literature, which led us to implement two significant improvements:
First, we refined our basal cell classification through detailed re-examination of our data. This enhancement has enabled more meaningful comparisons with recent landmark studies:
1.    We have revised Figure 2B by conducting a higher-resolution analysis of the Basal population, which we now subdivide into proliferating basal , basal, differentiating basal, suprabasal and transitioning basal populations. This refinement expands our original five cell types (proliferating basal , basal, suprabasal, secretory and ciliated) to seven distinct populations. This enhanced classification has particularly facilitated direct comparison with study [14], especially regarding the newly identified TAB population.
2.    We have added Figure S4 to demonstrate the comparison between our refined basal subtype, transitioning basal, and the TAB population reported in recent literature, strengthening the validation of our findings.
Second, addressing this comment in conjunction with Comments 3 and 4 from another reviewer, we have substantially expanded our comparative analysis of ciliated cells:
1.    We have enhanced the discussion section (page 13, lines 444-454) with comprehensive comparisons to existing literature regarding ciliated cell development and differentiation patterns in lung organoids.
2.    This addition provides valuable context for understanding both the commonalities and unique aspects of our findings relative to previous studies.
These revisions collectively strengthen our manuscript by providing thorough comparative analyses while highlighting the enhanced resolution of our cell type classification.

Comments 9: Please revise the English writing throughout the whole manuscript.
Response 9: Thank you for this important feedback. We have had the entire manuscript professionally edited by a scientific editing service to ensure clarity and accuracy of the English writing. Attached is the manuscript editing certificate provided by a specialized company. 
Following their guidance, we made some minor modifications to several figures. As recommended, we added axis labels to all UMAP plots, designating UMAP1 as the X-axis and UMAP2 as the Y-axis. 

We sincerely appreciate your thorough and insightful review, which has significantly enhanced the quality and depth of our manuscript. Your constructive feedback has helped us identify important areas for improvement and has guided us in providing a more comprehensive analysis of our findings within the broader context of the field. These revisions have strengthened both the scientific rigor and clarity of our work.

Round 2

Reviewer 1 Report

Comments and Suggestions for Authors

The age of participants is crucial, as the number of ciliated cells decreases with age and impacts the number of lung progenitor cells. Therefore, it is necessary to report the ages of the patients involved in the study. Specific ages can be anonymized by reporting them within a 5-year range, such as 30–35 years or 85–90 years. Additionally, smoking, chronic obstructive pulmonary disease (COPD), and asthma are associated with alterations in the number of ciliated cells in the lungs. The authors should also verify whether patients had any history of these conditions and report accordingly.

This study included only two donors and lacked technical replicates for the scRNA-seq experiments. Importantly, the small sample size imposes limitations on the findings of this study and should be mentioned in the discussion.

Author Response

Dear Reviewer,

We are truly grateful for the thoughtful review you have provided, and we have thoroughly reviewed all the comments and revised the manuscript accordingly. Your insights have been essential in strengthening our work and enhanced the manuscript.

Please find our responses to your comments below, with all corresponding updates highlighted in the manuscript.

Comments 1:  The age of participants is crucial, as the number of ciliated cells decreases with age and impacts the number of lung progenitor cells. Therefore, it is necessary to report the ages of the patients involved in the study. Specific ages can be anonymized by reporting them within a 5-year range, such as 30–35 years or 85–90 years. Additionally, smoking, chronic obstructive pulmonary disease (COPD), and asthma are associated with alterations in the number of ciliated cells in the lungs. The authors should also verify whether patients had any history of these conditions and report accordingly.
Response 1: We appreciate these important points of clarification. We have added the age, smoking information in the revised manuscript (page 2, lines 79--80) and Figure 1 (page 5). Additionally, as both patients had no history of respiratory disease, which we have clarified this in the manuscript (page 2, lines 79--80).

Comments 2: This study included only two donors and lacked technical replicates for the scRNA-seq experiments. Importantly, the small sample size imposes limitations on the findings of this study and should be mentioned in the discussion.
Response 2: We fully agree with this and thank you for this important feedback. We have added this limitation to the Discussion section (page 15, lines 503--page 16, lines 506).

Once again, we are deeply grateful for the valuable feedback from you. These revisions have strengthened the overall scientific rigor of our work, and we hope the manuscript now meets your expectations.

Reviewer 2 Report

Comments and Suggestions for Authors

No

Comments on the Quality of English Language

No

Author Response

Dear Reviewer,

Thank you for your previous valuable advice, which has significantly improved our manuscript.

In response to your feedback under the "Quality of English Language" section, indicating that "The English could be improved to more clearly express the research," we have had the entire manuscript professionally edited by a certified translation agency. The certificate of proofreading is attached for your reference.

Also, we carefully reviewed the entire manuscript again and revised awkward expressions and word choices. The revised sections are highlighted in red in the manuscript file. In particular, we have made the expressions on page 4 (lines 168–169), page 8 (lines 281–282), and page 14 (lines 410–411) more concise and clear.

Thanks to your feedback, we also discovered and corrected several errors that we had previously overlooked, such as on page 4 (lines 155–156), page 10 (lines 320–321), and page 14 (line 400).

We sincerely appreciate your time and effort in reviewing our manuscript. Your thoughtful input and feedback have been invaluable in refining our work.

Round 3

Reviewer 1 Report

Comments and Suggestions for Authors

Lines 80-81: The description of the patient is a little strange. Consider changing it to: "Both donors were in the 55-60 and 70-75 age brackets, and were a smoker and a non-smoker, respectively." Or the editorial staff may have a good suggestion.